# Validating the Five-Item World Health Organization Well-Being Index

**DOI:** 10.3390/ijerph191811489

**Published:** 2022-09-13

**Authors:** Mats Nylén-Eriksen, Ann Kristin Bjørnnes, Hege Hafstad, Irene Lie, Ellen Karine Grov, Mariela Loreto Lara-Cabrera

**Affiliations:** 1Institute of Nursing and Health Promotion, Oslo Metropolitan University, 0130 Oslo, Norway; 2Vårres Regional User-Led Center Mid-Norway, 7010 Trondheim, Norway or; 3Center for Patient-Centered Heart and Lung Research, Department of Cardiothoracic Surgery, Division of Cardiovascular and Pulmonary Diseases, Oslo University Hospital, 0424 Oslo, Norway; 4Department of Health Sciences, Faculty of Medicine and Health Sciences, Norwegian University of Science and Technology (NTNU), 2815 Gjøvik, Norway; 5Department of Mental Health, Faculty of Medicine and Health Sciences, Norwegian University of Science and Technology (NTNU), 7491 Trondheim, Norway; 6Nidelv Community Mental Health Centre, Division of Psychiatry, St. Olav’s University Hospital, 7006 Trondheim, Norway

**Keywords:** COVID-19, caregivers, psychological distress, psychometric properties, validity, well-being, WHO-5

## Abstract

***Purpose:*** Research on the psychological well-being of caregivers of children diagnosed with Attention-Deficit/Hyperactivity Disorder (ADHD) suggests that the well-being of parents and caregivers has been negatively affected by the COVID-19 pandemic. Although the psychological well-being of caregivers is a major concern, few validated well-being measures exist for caregivers of children diagnosed with ADHD. Therefore, a valid self-report scale is needed to assess well-being during the pandemic. The brief Five-Item World Health Organization Well-Being Index (WHO-5) has previously been used in studies on caregivers. However, its validity in this population remains unknown. This study aimed to evaluate the reliability and construct validity of the WHO-5 with caregivers of children with ADHD. ***Methods***: A cross-sectional anonymous online survey was conducted in Norway. The study recruited caregivers from a community sample during the COVID-19 pandemic. This was carried out to investigate the construct validity by exploring the relationship between well-being, quality of life, social support, self-reported psychological distress, and perceived stress. ***Results***: The findings of unidimensionality and high internal consistency, together with the results from the hypothesis testing, demonstrate the reliability and construct validity of the Norwegian version of the WHO-5 in this population. ***Conclusions:*** This study provides the first empirical evidence of the validity and reliability of the WHO-5 from a sample of Norwegian caregivers of children diagnosed with ADHD, with excellent reliability and construct validity. The scale can be used to systematize the measurement of well-being in caregivers because of its brevity and good psychometric properties, making it a valuable resource in research settings and assisting healthcare professionals in their crucial work of caring for caregivers.

## 1. Introduction

The COVID-19 pandemic is affecting the psychological well-being of millions of people worldwide [1], posing a threat to families and caregivers [2,3]. Preliminary evidence suggests that the COVID-19 pandemic has negatively affected parents’ and caregivers’ mental health and well-being [4,5,6]. For instance, an Italian study examining the impact of COVID-19 on the well-being of parents and children reported that coping with quarantine was a stressful experience for parents who were required to balance personal life, work, and raising children while being left alone with no other resources [7]. Besides the disruption in routines, caring for children while working from home negatively impacts parents of children with chronic conditions [8]. In general, during the COVID-19 pandemic, a potential risk of increased psychological distress (i.e., depression, anxiety and stress) was associated with emergency measures, social distancing, and community lockdowns for parents and caregivers [8]. However, in parents of children with chronic mental conditions, stress levels, depression and anxiety were significantly higher compared to those with healthy children [8]. These preventive measures may be exacerbated by a lack of social support, which eliminates the source of support from colleagues, reduces structural support, and disrupts connection with family and friends [9]. Despite mounting evidence that the COVID-19 pandemic has detrimentally affected caregivers of children with neurodevelopmental disorder (NDD) [10,11], surprisingly few studies focus on the impact of the pandemic on the well-being of parents and caregivers of children with ADHD [2,5,12].

During the pandemic, changes in daily routine impacted children’s behavior, affected sleep quality, irritability, and oppositionality [13]. However, studies show that ADHD in children can negatively impact parents’ stress, quality of life, and psychological well-being in normal times [14,15,16]. It has been argued that since many children with ADHD require mental health services, caregivers suffer an additional burden when these services are disrupted, reduced, or cancelled [13]. The social lockdown and isolation scenario seems to have posed difficulties for parents. Moreover, parents of children with NDD reported significantly higher psychological distress levels [17] and were observed to have a higher prevalence of depressive symptoms and anxiety [18,19]. Increased levels of psychological distress, depressive symptoms, and anxiety, especially during a pandemic, might impair their parenting abilities and behaviors, which in turn will aggravate their children’s challenges [20], leading to a negative downward trajectory. Caregivers also perceived less social support than parents of children without disabilities during the pandemic [21]. Previous studies reported that social support benefits the psychological well-being of caregivers of children with neurodevelopmental disorders because it reduces stress [22].

In Norway, the COVID-19 pandemic may present additional challenges as a result of infection control measures implemented to prevent the spread of COVID-19, most notably homeschooling, working from home, and the partial lockdown of mental health services. Caregivers of children with ADHD must assume a significantly increased responsibility during lockdown as schools are closed, supportive services have reduced facilities, and out-of-home leisure time activities are either diminished or cancelled [23]. It is reasonable to believe that these challenges, combined with the restrictions imposed by the COVID-19 pandemic and the stress of working from home, may endanger caregivers’ psychological well-being. However, more research is needed to better understand the impact of COVID-19 on caregiver’s well-being in this unique situation.

Despite recognizing the importance of caregivers’ well-being during these challenging times, few robust psychometrically tested instruments are available to assess it. Considering that healthcare professionals play a key role in caring for and supporting the caregivers and parents of children with mental and neurodevelopmental disorders, especially during the COVID-19 pandemic, they should be equipped with the necessary tools to fulfil their role; therefore, reliable, brief, and simple-to-use well-being instruments are required. The WHO-5 is a widely used generic self-report instrument that measures psychological well-being, even though the original version was developed to measure depression in patients with diabetes [24]. Validation studies have been conducted in different countries and fields [24,25]. The instrument was conceptualized as a unidimensional measure in the diabetes setting, and studies in mental health settings, both in adolescent [26,27] and adult [28,29,30] populations, support its validity. Still, there is currently limited evidence on the psychometric properties of the WHO-5 for caregivers in general, and especially in caregivers of children with NDDs such as ADHD [24,25]. Having been validated in patient populations, in both somatic and mental health settings, and in the general public, we argue that validation with caregivers is warranted as this population represents neither a patient population nor the general public. When compared with the general public, caregivers of children with NDDs such as ADHD are associated with higher levels of depression [31]. Caregivers and their families have an increased risk of mental health challenges and decreased family well-being [32,33]. Seeing that the caregiver population differs from the already validated populations, a validation of the WHO-5 in this population is warranted. In order to provide the best possible health services for caregivers of children with ADHD, the limited evidence on the psychometric properties of the WHO-5 in this population needs to be amended. Amending the limited evidence will additionally ensure that healthcare workers use the appropriate instrument and base their evaluations and the conclusions drawn on valid and reliable data. As a healthy family life and environment is essential to support child development [33], validating the WHO-5 in this population seems important not only for caregivers, but also for their children.

### The Present Study

To meet the demand for further evidence in this critical research area, we aimed to investigate the factorial structure, internal consistency, and construct validity of the questionnaire in this present study. The hypotheses used to explore the construct validity are presented in the methods section and are based on the existing knowledge on the topic.

## 2. Methods

### 2.1. Data Source and Procedures

Data were gathered in a cross-sectional study conducted from 9 June to 30 June 2020, and we refer to this sample as the COVID-19 sample. In this anonymous survey, we recruited caregivers from the Norwegian Association for ADHD who reported living with children. Inclusion criteria for caregivers in this study were: (1) aged 18 years and older; (2) proficient in spoken Norwegian; (3) able to provide informed consent; and (4) being a caregiver living with a child/children diagnosed with ADHD. Before carrying out the survey, the WHO-5 was revised and piloted by user representatives from the user organization Vårres Regional User-led Center Mid-Norway. Three user representatives examined the content validity of the items, reviewed their relevance, and provided feedback to ensure language representation for ease of understanding. The survey was distributed via email by the study collaborators to 496 potential participants. The emails provided information about the study, the name of the responsible investigator, and an electronic link to the questionnaire. Possible participants were able to forward the online survey web link to other networks and via social media. Confidentiality and anonymity were maintained by not asking for names or other direct identifiers that could connect the data to the individual who provided it, such as addresses or other identification. No identification list was created, and a completed survey generated the data for anonymous storage. By clicking “I agree”, the participants indicated that they had read the consent form and agreed to participate in the research study. The survey settings were set to refuse multiple responses from the same IP address. Once the survey was retrieved from the survey software providers, the researcher created an anonymous SPSS-file protected with a two-factor authentication login system. The study was conducted in accordance with the Declaration of Helsinki [34]. The self-administered questionnaires required 15 min to complete using the Questback software. The first page of the online survey provided an information sheet outlining the study purpose and use of the data. The study was planned to investigate the construct validity of the WHO-5 via hypothesis testing. This was conducted by investigating the relationship between well-being (WHO-5), quality of life (MQLI), social support (OSSS-3), perceived stress (PSS-4), and anxiety/depression/psychological distress (PHQ-4). For each of the elements included in the investigation, a self-reported questionnaire was administrated.

### 2.2. Measurements

#### 2.2.1. The Five-Item World Health Organization 5-Item Well-Being Index (WHO-5)

The WHO-5 is a generic, self-reported instrument that includes the following five easy to understand Likert-type statements to evaluate psychological well-being [24]: “I have felt cheerful and in good spirits”, “I have felt calm and relaxed”, “I have felt active and vigorous”, “I woke up feeling fresh and rested” and “My daily life has been filled with things that interest me”. Caregivers were asked to rate their agreement over the previous two weeks on each of the items rated on a 6-point scale from “all of the time” to “at no time”. The internal consistency was reportedly good in previous validation research conducted in mental health settings, with Cronbach’s coefficient ranging from 0.83 to 0.92 [30].

#### 2.2.2. Three-Item Oslo Social Support Scale 3 Items (OSSS-3)

The OSSS-3 [35] is a brief questionnaire that assesses levels of social support. It consists of only three items that ask for the number of close confidants, the sense of concern from other people and the relationship with neighbors with a focus on the accessibility of practical help. It has been applied in several large-scale population-based surveys in different settings, and a Cronbach’s α of 0.640 has been reported [36].

#### 2.2.3. Perceived Stress Scale Four Items (PSS-4)

The PSS-4 [37] assesses perceptions of stressors and how frequently they occur. The PSS has been classified as a reliable and valid self-reporting measure among students [37,38,39,40]. A Cronbach’s α of 0.720 has been reported [38]. The instrument includes four items, and each item uses a 5-point scale ranging from “never occurred” to “very often”.

#### 2.2.4. Patient Health Questionnaire for Depression and Anxiety (PHQ-4)

PHQ-4 is a validated [41], ultra-brief tool used for detecting both depression and anxiety with a Cronbach’s α of 0.850. The four-item self-rated questionnaire is a combination of the validated two-item ultra-brief screeners for Depression (PHQ-2) and Anxiety (GAD-2). The respondents are asked “Over the *last 2 weeks,* how often have you been bothered by the following problems” and then to score each item on a 4-point scale ranging from “not at all” to “nearly every day” [41].

#### 2.2.5. Three-Item Multicultural Quality of Life (MQLI-3)

The Short MQLI is a 3-item self-rated questionnaire that assesses 3 dimensions of the concept of quality of life including physical well-being, self-care and independent functioning and occupational functioning. Each item is rated on a scale from 1 to 10 according to the subject’s culture-informed understanding of the concept [42]. The MQLI-3 has been validated in several populations including patients, students, and healthcare professionals [42,43,44,45,46,47,48] and the Cronbach’s α is documented to range from 0.730 [48] to 0.920 [42].

### 2.3. Ethical Considerations

The study was conducted after receiving approval from the Regional Committee for Medicine and Health Research Ethics in Mid-Norway (ref.: 2020/149184). Participants were provided with a study information sheet and were informed that they could omit items and discontinue the survey at any time. Consent was implicitly provided by anonymously responding to the questionnaires and returning the answers.

### 2.4. Statistical Analyses

A data analysis was carried out using the Statistical Program for Social Sciences (SPSS IBM Corp., Armonk, NY, USA), version 27 and Mplus (Muthén & Muthén, Los Angeles, CA, USA), version 8.4. To describe the sample, the count (n), means, standard deviation (SD), frequencies, and percentages were calculated. To evaluate acceptability we calculated missing data, floor, and ceiling effects at item levels. A percentage of greater than 15% indicates the presence of a floor or ceiling effect [49].

The psychometric evaluation included internal consistency and construct validity. WHO-5 was characterized using the mean, transformed to 0–100 (higher scores indicate better well-being). All the psychometric analyses used raw scores. The internal consistency of the scale was investigated by computing Cronbach’s alpha and McDonald’s Omega. A Cronbach’s alpha of 0.7 or higher represents acceptable consistency [49]. Prior to performing the exploratory principal component analysis (PCA), the data were tested for suitability of data for a factor analysis based on sample size, factorability of the correlation matrix (at least some correlations of *r* = 0.3 or greater; Bartlett’s test of sphericity (*p* < 0.05); and Kaiser–Meyer–Olkin (≥0.6)), linearity and outliers [50]. All five items of the WHO-5 were included in the exploratory PCA to analyze the construct validity and component structure. The factor extraction (retaining components) was based on the Kaiser criterion (eigenvalue ≥ 1) and inspection of the scree plot [50]. A confirmatory factor analysis (CFA) was subsequently conducted by performing structural equation modeling to confirm the factor structure resulting from the PCA. The Chi-square X^2^, comparative fit index (CFI), Tucker–Lewis index (TLI) and root mean square error of approximation (RMSEA) were calculated. Acceptable threshold levels for the different fit indices are as follows: low X^2^ relative to degrees of freedom with *p* > 0.05, CFI/TLI > 0.95 and RMSEA < 0.07 [51]. All five items of the WHO-5 were included in the CFA.

The construct validity was explored using hypothesis testing. Previous studies [52,53] indicated that social support is associated with well-being, so we expected a moderate positive correlation between the OSSS-3 and WHO-5. We also hypothesized, consistent with prior research, that perceived stress and anxiety/depression would be negatively correlated with well-being (WHO-5) [54,55,56] while quality of life (MQLI-3) would be positively correlated with well-being [48]. Spearman’s correlations were adopted for non-parametric variables after testing for normality [50].

## 3. Results

A total of 213 caregivers of children with ADHD, 162 mothers (76.1%) and 50 fathers (23.5%) completed the questionnaires including WHO-5, OSSS-3, PSS-4, PHQ-4, and MQLI-3. There were no floor/ceiling effects, and the missing values were <1%. The socio-demographic characteristics of the sample are presented in Table 1. The descriptive statistics of the items in the WHO-5, including factor loading, is presented in Table 2. The average well-being score was 46.48. The overall Cronbach’s α and McDonald’s omega (ω) for the OSSS-3, PSS-4, PHQ-4, and MQLI-3 were, respectively; 0.672, 0.788 (0.7957), 0.855 (0.8541), and 0.763 (0.5729). Hypothesis testing exploring the construct validity is presented in Table 3.

### 3.1. Internal Consistency

The Norwegian translated WHO-5 scored a relatively high internal consistency (Cronbach’s α coefficient) of 0.875, and (McDonald’s ω coefficient) of 0.8796, thereby representing acceptable consistency which indicates that the instrument measures only one concept. All items in the inter-item correlation matrix were positive and the corrected item-total correlation ranged from 0.635 to 0.762.

### 3.2. Factor Analysis of WHO-5

The five items of the WHO-5 were subjected to PCA. The suitability of the data for factor analysis was assessed prior to performing the PCA. The sample size meets the criterion of 150+ and there was a ratio of at least five cases for each of the variables [50]. The correlation matrix presented coefficients from 0.532 to 0.676. The Kaiser–Meyer–Olkin value of 0.865 and a significant (*p <* 0.001) Bartlett’s Test of Sphericity support the appropriateness of the factor analysis. There was no clear evidence of a curvilinear relationship and no outliers among the cases.

The PCA suggested an extraction of one factor from the five items of the Norwegian-translated WHO-5 by revealing the presence of one component with the eigenvalue meeting the Kaiser’s criterion > 1, explaining 67.4% of the variance. Inspection of the scree plot generated by the PCA supported the one-factor finding. The CFA confirmed and supported the one-factor structure of the Norwegian-translated WHO-5, with X^2^ = 9.222(5), *p* = 0.1005, an RMSEA of 0.063, CFI of 0.988 and TLI of 0.976. Figure 1 presents the factor structure, factor loadings and residual variance for each item of the WHO-5 from the CFA.

### 3.3. Hypothesis Testing

The correlation test between the WHO-5 and OSSS-3 found a moderate correlation and a strong correlation between the WHO-5 and MQLI-3 as hypothesized. Similarly, we found a strong negative correlation between the WHO-5, PSS-4 and PHQ-4 as hypothesized (Table 3).

## 4. Discussion

To the best of our knowledge, the caregiver population is the only group in which the WHO-5 has not been validated, and as this group differs from patient populations, in both somatic and mental health settings, and the general public, a validation was warranted. The ongoing COVID-19 pandemic may negatively impact the psychological well-being of this vulnerable population, necessitating the use of a validated instrument to easily identify those in need. To address this critical need, the psychometric properties of the Norwegian version of the WHO-5 were investigated. The findings of the present study, which is the first to investigate the factor structure and reliability of the WHO-5 in a sample consisting of caregivers of children with ADHD during the COVID-19 pandemic, show high internal consistency reflecting the concept of well-being and one-factor structure. One component suggesting the extraction of one factor from the WHO-5 by the PCA-test was found, with Eigenvalues explaining 67.4% of the variance and further supported by the generated scree plot. This was further confirmed and supported by the CFA model reaching acceptable threshold levels for chi-square X^2^, CFI/TLI and RMSEA, indicating good model data fit [51]. The findings of unidimensionality [24] and high internal consistency [28,30], together with the results from the hypothesis testing, demonstrate the reliability and construct validity of the Norwegian version of the WHO-5 in this population. In the present study, four pre-defined hypotheses were created to test construct validity, and the analysis resulted in rejection of the null hypotheses for all four hypotheses (Table 3).

Our result indicated a moderate positive correlation between social support (OSSS-3) and well-being (WHO-5). The positive correlation between these concepts supports the construct validity of the WHO-5, and Simeon et al. [52] found a similar moderate positive correlation between social support and well-being in 550 Austrian citizens (74% female, mean age 40.22 years) at the beginning of the COVID-19 lockdown. Additionally, poor well-being was associated with negative capability and worse mental health. Social support is shown to impact well-being, and in a UK community sample, community identification and well-being were mediated by increased social support and reduced loneliness [57]. Similarly, the results of the current study regarding social support and well-being are in accordance with a smaller, cross-sectional study of 81 parents of children with neurodevelopmental disorders, in which the parents, independent of their own ADHD status, reported functional impairments related to spouse or partner relationship and social functioning, and the negative association was even stronger among parents with a positive ADHD status [58]. The study supported our hypothesis that social support impacts caregiver well-being as demonstrated in another larger US sample (N = 7565). Worse caregiver well-being was significantly associated with ADHD symptoms in preschool-age children and fully mediated the relationship between social determinants of health and ADHD symptoms [59].

As hypothesized (Table 3), our results revealed strong negative correlations for both perceived stress (PSS-4) and psychological distress (PHQ-4) with well-being (WHO-5), i.e., correlation coefficients (Spearman’s rho) of −0.706 and −0.736, respectively, demonstrating evidence of the construct validity of the WHO-5. These results are comparable to a study of 214 nurses (female 77%, mean age 40.3 years) who provided direct care during the COVID-19 pandemic in Tenerife [55]. Similar negative correlations were found in a longitudinal cohort study during the first wave of the COVID-19 pandemic of parents of 159 children with externalizing difficulties, in which the parents experienced significantly high stress levels and low levels of well-being [56]. A study validating the Brazilian Portuguese version of the WHO-5 shows that higher well-being scores (WHO-5) were negatively correlated with symptoms of depression [60]. The negative correlation between depression symptoms and well-being was expected, as the WHO-5 is also validated as a screening tool for depression as well as a generic instrument for well-being [24,28]. This correlation demonstrated evidence of convergent validity. Based on the above-mentioned knowledge of the mental health of caregivers of children with ADHD during the pandemic, including perceived stress, psychological distress, and the strong negative correlations with well-being, it is apparent that a critical need exists for a tool such as the WHO-5 to assist healthcare professionals in their key role of caring for and supporting this population.

Finally, we hypothesized that there would be a strong positive correlation between quality of life (MQLI-3) and well-being (WHO-5), which was revealed by our results; as the correlation coefficient (Spearman’s rho = 0.730) shows, there is a strong positive association between these two instruments. Normally, in construction studies in which these two instruments are suggested, the researchers must be aware of the conceptual overlap and consider this when using them as dependent and independent variables in a model. Items with high multicollinearity should be removed from the model in order to avoid tautology. However, in this study, these two instruments were purposefully included to investigate the construct validity, i.e., the convergent validity of the WHO-5, and the results of the current study support the idea of the convergent validity of the WHO-5. These findings are consistent with other studies [48]. According to the systematic review and meta-analysis on the psychological and behavioral impact of lockdown and quarantine measures of the COVID-19 pandemic on children, adolescents, and caregivers [18], of the 257 caregivers in the study 52.3% developed anxiety, and 27.4% developed depression during isolation with their children. Compared with those of previously healthy children, the parents of children with behavioral co-morbidities such as ADHD perceived a significantly higher need for professional support during the pandemic [18]. Thus, it was not surprising that the mean score on the WHO-5 was found to be <50 (46.48), which according to Topp et al. [24] is the recommended cut-off when using the WHO-5 for depression screening. However, Panda et al. [18] found that with appropriate support and assistance, the parents of children with ADHD can use isolation with their children as an opportunity and benefit the family; furthermore, the children might even experience an improvement in their symptoms. Still, for the caregiver to identify an opportunity rather than a crisis, they need appropriate support.

The key roles of healthcare professionals are to see, support, help and care for the caregivers in this population, and the importance of having the best tools for the job has become increasingly apparent in this area. The current COVID-19 pandemic affects the well-being of children with ADHD and their caregivers. Leadership from healthcare professionals is needed to safeguard the quality of care. Healthcare professionals should collaborate to initiate advanced care, plan conversations on time, review and document improved care plans, and adapt goals of care as needed in light of the COVID-19 situation. It can be difficult for a healthcare professional to determine with certainty whether a caregiver is struggling with their mental health or well-being, especially during the pandemic. Healthcare workers need time to achieve this appropriately. The WHO-5, which is short and easy to use, is suitable for healthcare professionals to map the subjective well-being of caregivers even when pressed for time [24]. Thus, to assist healthcare professionals in this crucial work, this current study investigated the Norwegian version of the WHO-5 and suggests that it is a psychometrically robust and validated instrument which through its simplicity, effectiveness and generic design can be used to screen for depression and serve as a generic instrument to assess the well-being of caregivers of children with neurodevelopmental disorders such as ADHD.

### Strength and Limitations

It is considered a strength of this study that, to the best of our knowledge, it is the first to validate the WHO-5 in this population during the ongoing COVID-19 pandemic. Our findings contribute to the future work on caring for caregivers, especially during troubling times such as the COVID-19 pandemic. In addition, the current study has the potential to contribute to the field by applying the WHO-5 tool in other situations in which parents and caregivers face extraordinary challenges in caring for children with neurodevelopmental conditions.

Because of the anonymous nature of the design, the study has limitations in terms of documenting respondents’ backgrounds, those who did not complete the survey, and the generalizability of the findings. It was challenging to find similar studies to compare our findings with, demonstrating the need for our current study. However, we found relevant studies providing the necessary bases to compare the findings.

## 5. Conclusions

This study provides the first empirical evidence of the validity and reliability of the WHO-5 from a sample of Norwegian caregivers of children diagnosed with ADHD, with excellent reliability and construct validity. The study offers new insight into the impact of COVID-19, highlighting the concerns regarding the psychological well-being of this caregiver population and providing a future focus. We recommend using the WHO-5 to measure well-being in caregivers due to its brevity and good psychometric properties, which make it a valuable resource in research settings and in assisting healthcare professionals in their crucial work of caring for caregivers.

## Figures and Tables

**Figure 1 ijerph-19-11489-f001:**
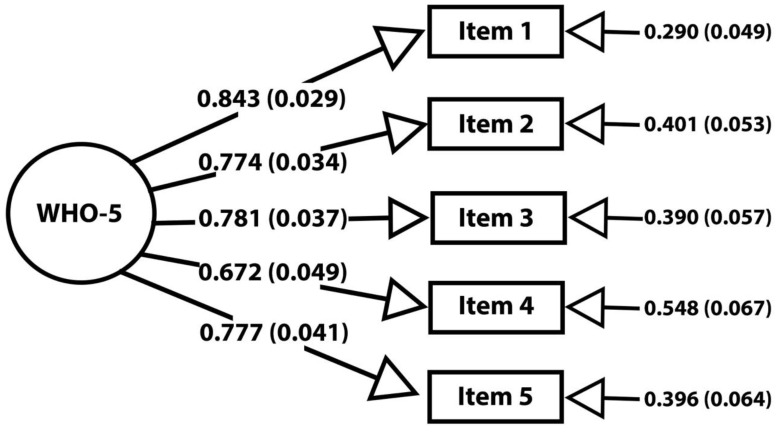
Structural equation model (SEM) for the WHO-5. Numbers on the arrow pointing from latent variable (WHO-5) to the items are the standardized factor loadings (with standard errors). The numbers pointing to the items from the right are the residual variance (with standard errors).

**Table 1 ijerph-19-11489-t001:** Socio-demographic characteristics of caregivers (N = 213).

	n (%)
**Sex**	
Woman	162 (76.1)
Man	50 (23.5)
**Age**	
18–29	15 (7.0)
30–39	72 (32.8)
40–49	96 (45.1)
50–≥65	29 (13.6)
**Marital status**	
Unmarried	37 (17.4)
Married/Cohabitant	149 (70.0)
Divorced/Separated	24 (11.3)
Widow/Widower	1 (0.5)
**Education**	
Primary and high school	99 (46.5)
University 3 years or less	80 (37.6)
University 5 years or more	34 (16.0)
**Working status**	
Student	14 (6.6)
Paid work/self employed	118 (55.4)
Sick leave	28 (13.1)
Administrative leave	1 (0.5)
Other	51 (23.9)

**Table 2 ijerph-19-11489-t002:** Descriptive statistics of the items in WHO-5.

	Item	Mean	Std. Deviation	Factor Loading
1	I have felt cheerful and in good spirit	2.83	1.134	0.863
2	I have felt calm and relaxed	2.30	1.174	0.821
3	I have felt active and vigorous	2.27	1.307	0.838
4	I woke up feeling fresh and rested	1.77	1.397	0.758
5	My daily life has been filled with things that interest me	2.46	1.234	0.823

Factor loadings from the PCA. WHO-5, Five-Item World Health Organization Well-Being Index.

**Table 3 ijerph-19-11489-t003:** Summarizing the hypothesis testing.

	Null Hypothesis	Test	Correlation Coefficient	Sig.	Decision
1	There is no positive correlation between social support (OSSS-3) and well-being (WHO-5)	Spearman’s rho WHO-5 OSSS-3	0.413	<0.001	Reject the null hypothesis
2	There is no negative correlation between perceived stress (PSS-4) and well-being (WHO-5)	Spearman’s rho WHO-5 PSS-4	−0.706	<0.001	Reject the null hypothesis
3	There is no negative correlation between psychological distress (PHQ-4) and well-being (WHO-5)	Spearman’s rho WHO-5 PHQ-4	−0.736	<0.001	Reject the null hypothesis
4	There is no positive correlation between quality of life (MQLI-3) and well-being (WHO-5)	Spearman’s rho WHO-5 MQLI-3	0.730	<0.001	Reject the null hypothesis

The *p*-values originate from Spearman’s rho (2-tailed). The correlations were conducted with sum scores. WHO-5, Five-Item World Health Organization Well-Being Index.

## Data Availability

Data and materials are available on request.

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
