# Peer review of "Validating the Five-Item World Health Organization Well-Being Index"

_ijerph, 2022, doi:10.3390/ijerph191811489_

Round 1

Reviewer 1 Report

I enjoyed reading this paper.  It is well written, relevant and timely.  I have a few minor suggestions below:

Line 131:  Can you speculate as to how many of the surveys were forwarded?  What percent of surveys completed were from your selection vs forwarded?  Do you think this could affect your results?

Line 147:  I was confused about all of the survey instruments being used until later in the paper when it was explained that each survey was administered.  Might that be better explained earlier in this section?

Line 148:  I like that the WHO-5 uses a 6 point Likert scale.  This should reduce the central tendancy bias.

Line 162: .640 seems on the low end of an acceptable Cronbach’s Alpha.  Did you have concerns about this?

Table 3: .413 Spearman’s Rho seems like a weak correlation.  Why do you think this is?

Line 287: You begin this sentence with “As we hypothesized”.  It might be helpful to the reader to formally state your hypotheses in full earlier in this section.

Reviewer 2 Report

I feel that the paper, ‘Validating the Five-Item World Health Organization Well-Being Index’ is well conceived and well written.  While it addresses a fairly esoteric social science issue – the validity of the WHO-5 assessment for parents and caregivers of children with ADHD – the methodology employed is very sound and the conclusions reached are well supported by the results.

One could argue that the pandemic-related stresses cited for such caregivers are primarily related to the period of the lockdown and are therefore no longer applicable, as Norway and the world have moved to a more nuanced approach to managing COVID-19.  I believe, however, that this validation of the WHO-5 assessment can now be applied in other situations in which parents/caregivers face extraordinary challenges in caring for children with neuro-developmental conditions.

Author Response

Please see attachement.

Reviewer 3 Report

Dear Authors,

first of all, I would like to congratulate you on the conducted research and emphasize its extreme relevance and importance, in particular from the point of view of practical application. I was extremely pleased to read the article, I am impressed with the clear structure of the article and the well-thought-out methods part. I am convinced that the results presented in the article will become the basis for further research.

At the same time, I would like to clarify some details that can add more clarity and novelty to the article and become useful for readers.

1. Does the linguistic validation of the questionnaire was provided (translation of the questionnaire into the Norwegian language)? Or perhaps this was done in previous studies?

2. Since the factor structure of the questionnaire is known according to previous studies, why as the analysis method EFA (not CFA) was chosen?
